# Laser Acupuncture versus Liraglutide in Treatment of Obesity: A Multi-Institutional Retrospective Cohort Study

**DOI:** 10.3390/healthcare12131279

**Published:** 2024-06-26

**Authors:** Wen-Lin Yu, Yu-Ning Liao, Tsung-Hsien Yang, Ching-Wei Yang, Ting-I Kao, Pai-Wei Lee, Chiu-Yi Hsu, Jhen-Ling Huang, Yu-Tung Huang, Hsing-Yu Chen

**Affiliations:** 1Division of Chinese Internal Medicine, Center for Traditional Chinese Medicine, Chang Gung Memorial Hospital, Taoyuan 333, Taiwan; mr3099@cgmh.org.tw (W.-L.Y.); x1991425@cgmh.org.tw (Y.-N.L.); 8905001@cgmh.org.tw (T.-H.Y.); ycw0426@gmail.com (C.-W.Y.);; 2School of Traditional Chinese Medicine, College of Medicine, Chang Gung University, Taoyuan 333, Taiwan; 3Center for Big Data Analytics and Statistics, Chang Gung Memorial Hospital, Linkou Medical Center, Taoyuan 333, Taiwan; wei.lee@parexel.com (P.-W.L.); joy960111@gmail.com (C.-Y.H.); cratwoy0309@gmail.com (J.-L.H.);; 4Graduate Institute of Clinical Medical Sciences, College of Medicine, Chang Gung University, Taoyuan 333, Taiwan

**Keywords:** laser acupuncture, integrative and complementary medicine, obesity, body weight loss

## Abstract

Background: Obesity is a global concern, driving the search for alternative treatments beyond lifestyle changes and medications. Laser acupuncture (LA) shows promise in obesity management, yet few studies compare it with FDA-approved medications. This study aimed to assess and compare LA’s impact with liraglutide on weight reduction in obese individuals. Methods: Data from the Chang Gung Research Database (CGRD) (2013–2018) were analyzed. Primary outcomes included changes in body weight and BMI within 180 days, with secondary outcomes measuring the proportion achieving 5%, 10%, and 15% weight loss. Adverse events were also assessed. Results: Of 745 subjects (173 LA users, 572 liraglutide users), LA users lost more weight by day 180 (5.82 ± 4.39 vs. 2.38 ± 5.75 kg; *p* < 0.001) and had a greater BMI reduction (−2.27 ± 1.73 vs. −0.93 ± 2.25 kg/m^2^; *p* < 0.001). More LA users achieved 5% and 10% weight loss compared to liraglutide users (64.2% vs. 22.7%, 26.6% vs. 4.2%; all *p* < 0.001). After balancing baseline differences, LA’s benefits remained significant. No adverse events were reported with LA. Conclusions: LA may offer superior weight reduction compared to liraglutide. Future studies should explore LA alone or in combination with liraglutide for obesity management.

## 1. Introduction

In modern society, obesity has become a worldwide problem due to high-calorie diets and sedentary lifestyles. The prevalence of obesity has been increasing in the past decades [1,2,3,4]. According to the definition established by the World Health Organization, individuals in the Asia–Pacific region with a body mass index (BMI) > 23 kg/m^2^ and >25 kg/m^2^ are deemed overweight and obese, respectively. Obesity has been linked to an increased risk of various diseases, including cardiovascular diseases [5,6], osteoarthritis, type 2 diabetes, dyslipidemia, polycystic ovary syndrome [7], non-alcoholic fatty liver disease, malignancies, obstructive sleep apnea, and depression [8,9,10]. Additionally, evidence has suggested that individuals who are overweight or obese are at a higher risk of mortality, and that these conditions are associated with a substantially increased burden on healthcare. Therefore, studies focusing on the management of obesity are warranted [1].

Currently, available therapeutic strategies for weight loss include dieting, lifestyle modifications, exercise, and the use of anti-obesity agents. Research indicates that exercise, diet control, medication, and surgery could result in small-to-modest reductions in body weight [11,12]. Lifestyle modification, including calorie limiting and regular physical activity, is the first-line treatment for alleviating obesity. However, maintaining these efforts over a long period can be challenging because of the busy modern lifestyle and poor execution ability [13,14]. Research indicates that consuming anthocyanins significantly reduces body weight, fat mass, and inflammation in murine obesity models, but human studies are lacking [15]. Surgery can be the most effective option for reducing body weight; however, the potential side effects of surgery (e.g., vitamin deficiency, gastroesophageal reflux, and dumping syndrome) remain a concern [12,16]. Therefore, medications for the management of obesity have become a mainstream approach to body weight reduction. The Food and Drug Administration (FDA) has approved orlistat, phentermine, phentermine/topiramate extended-release, naltrexone/bupropion, and two glucagon-like peptide-1 (GLP-1) agonists (liraglutide and semaglutide) for weight control [16]. Liraglutide promotes weight loss by inducing the feeling of fullness and delaying gastric emptying, leading to decreased food intake [17]. According to the SCALE trial, among patients with diabetes mellitus (DM), 15.9% of liraglutide users lost 10% of their body weight [18]. Moreover, liraglutide reduced body weight in obese individuals without DM [19]. Nevertheless, the occurrence of adverse events (e.g., nausea and elevated risk of cholelithiasis, cholecystitis, and pancreatitis) may limit the use of liraglutide [20]. 

In addition to Western medicine, the use of complementary and alternative medicine (CAM) for obesity management currently has numerous adherents. This is primarily attributed to intolerance to Western medicine or concerns regarding side effects [21]. Several studies reported that laser acupuncture (LA) and diet control reduced the waist-to-hip ratio, BMI, and appetite in obese individuals compared with sham treatment [22,23,24]. A systematic review also indicated that LA might positively affect obesity by reducing body weight, body fat, and appetite [25]. However, due to the small sample sizes and relatively short treatment duration, it is difficult to draw a robust conclusion regarding using LA to alleviate obesity. Additionally, most studies compared the effects of LA to those of placebo or sham treatment. Hence, it remains unclear whether the effectiveness of LA in reducing body weight is comparable to that of FDA-approved medications.

This real-world cohort study aimed to compare the effects of LA and liraglutide on body weight reduction among obese individuals. By obtaining results from real-world data, we can evaluate the effectiveness of LA in daily clinical practice before conducting clinical trials in the future.

## 2. Materials and Methods

### 2.1. Data Source

We collected data from the Chang Gung Research Database (CGRD), which records procedures, medications, laboratory data, and examinations from outpatient, inpatient, and emergency visits at the Chang Gung Memorial Hospital (CGMH). CGMH is the largest hospital system in Taiwan, with eight medical institutes, including 10,070 beds with >280,000 inpatient admissions, >8,500,000 visits to the outpatient department, and 500,000 visits to the emergency department annually [26]. The CGRD covers a significant portion of the Taiwanese population, accounting for 12.2% and 21.2% of inpatient and outpatient services, respectively, between 1997 and 2020 [27]. The enormous amount of available medical data renders the CGRD an excellent resource for clinical studies [11,26,28]. 

### 2.2. Study Design and Subjects

We collected all feasible patient data from 1 January 2013 to 31 December 2018. Patients diagnosed with obesity or metabolic syndrome, confirmed by the International Classification of Diseases, Tenth Revision and Ninth Revision, Clinical Modification (ICD-10-CM and ICD-9-CM) codes, were selected. These codes included ICD-9-CM codes 278.00, 278.01, 278.03, 649.10-14, 793.91, V85.30-39, V85.41-45, and V85.54, and ICD-10-CM codes E65, E668, and E669. The first day of LA or liraglutide use was set as the index date. The baseline body weight was determined as the highest body weight recorded one month before and after the index date, and the BMI was calculated using the height of each subject. 

The inclusion criteria were (1) age ≥ 18 and ≤75 years; (2) BMI ≥ 27 kg/m^2^ and at least one weight-related comorbidity, such as DM, dyslipidemia, hypertension, or a fatty liver 360 days before the index date; and (3) at least two treatments with LA or liraglutide after the index date. Subjects with missing data on weight or height at baseline or day 180, as well as those who used other anti-obesity medications or who had a history of bariatric surgery, were excluded (Figure 1). 

### 2.3. Intervention and Comparisons

The liraglutide group received subcutaneous liraglutide according to the recommended dosing scheme for obesity. The LA group received LA three to four times per week on fixed acupoints from a qualified practitioner of traditional Chinese medicine. The choice of acupoints and frequency of LA was based on the study conducted in 2010 by Dr. Wen-Long Hu [24]. The selected acupoints were as follows (Figure 2, Appendix A): stomach and hunger (auricular points; B6 for right ear and B7 for left ear), ST25 (Tianshu, B2), ST28 (Shuidao, B2), ST40 (Fenglong, B2), SP15 (Daheng, B2), and CV9 (Shuifen, B3) [24]. Energy (0.5 J) was applied to each of the above points via a gallium aluminum arsenide Handylaser Trion (RJ-Laser; Reimers & Janssen GmbH, Waldkirch, Germany) with the following settings: maximal power, 50 mW; wavelength, 785 nm; area of probe, 0.03 cm^2^; power density, 50 mW/cm^2^.

### 2.4. Covariates

Covariates including age, gender, height, body weight, mean arterial pressure, and underlying comorbidities (e.g., hypertension, dyslipidemia, ischemic heart disease, cerebrovascular diseases, chronic pulmonary disease, DM, chronic hepatitis, non-alcoholic steatohepatitis) were collected. The Charlson comorbidity index (CCI) was also calculated to summarize the comorbidities (the diagnostic codes are listed in Appendix A). The use of diagnostic codes at least twice in outpatient visits or once during admissions six months before the index date denoted the occurrence of the abovementioned comorbidities. The worst value recorded within six months before the index date was used to retrieve biochemical profiles, including aspartate transaminase, alanine transaminase, blood urea nitrogen, creatinine, hemoglobin A1c, fasting sugar, total cholesterol, triglyceride, low-density lipoprotein, and high-density lipoprotein. 

### 2.5. Outcome Assessment

The primary outcome of this study was the change in body weight from baseline to day 180. The secondary outcome was the proportion of subjects who lost at least 5%, 10%, and 15% of their baseline weight. We also assessed the temporal changes in body weight at days 30, 60, 90, 120, and 150. Any newly diagnosed ICD codes used during admissions or emergency visits within the 180-day treatment course were regarded as adverse events, including hypertension, ischemic stroke, hemorrhagic stroke, and cardiovascular disorders (the diagnostic codes are listed in Appendix A).

### 2.6. Statistical Analysis

The baseline covariates were described, and inference statistics were performed to examine the differences between the LA and liraglutide. Furthermore, overlap weighting was used to overcome the potential confounding bias caused by differences in demographic features and imbalance in LA and liraglutide users. Overlap weighting is a statistical method commonly used in observational studies that rely on propensity scores (PS) to mimic the randomization process of clinical trials, especially when there are imbalances between groups [29]. First, baseline covariates, including age, gender, body weight, and CCI, were used to generate the PS for using LA. Second, the values of 1-PS and PS were assigned to the LA group and liraglutide group, respectively, as weights when estimating the outcomes. The absolute standardized mean difference (ASMD) was used to demonstrate the differences between LA and liraglutide groups with overlap weighting. Sensitivity tests were conducted with other PS-related models, such as 1:1 propensity score matching (PSM) and inverse probability of treatment weighting (IPTW) with the same PS value as overlap weighting to verify the consistency of the results. The differences in weight change between LA and liraglutide users at each time point were examined using independent *t*-tests. Pearson *X*^2^ statistics were used to compare the proportions of subjects who achieved 5%, 10%, and 15% weight reduction, as well as the rates of adverse events between the two groups. Repeated ANOVA was used to examine the treatment effects within and between the LA and liraglutide users. The effect size was calculated with a 95% confidence interval (CI) for proportional data. Statistical calculations were performed using the commercially available software SAS Studio 3.4 (SAS, Cary, NC, USA), with *p*-values < 0.05 denoting statistically significant differences, and Bonferroni correction was used when multiple comparisons were performed.

## 3. Results

### 3.1. Baseline Characteristics of Subjects

A total of 745 eligible subjects were included in the analysis, of which 173 (23.2%) were LA users. The baseline characteristics of these two groups are summarized in Table 1. Compared with liraglutide users, LA users were younger (51.7 ± 11.7 vs. 41.7 ± 11.0 years, respectively) and predominantly female (50.7% vs. 83.8%, respectively; *p* < 0.001). In addition, LA users had lower baseline body weight than liraglutide users (81.4 ± 13.1 vs. 87.1 ± 16.9 kg, respectively; *p* < 0.001). Moreover, compared with liraglutide users, LA users had a lower prevalence of comorbidities, including hypertension (61.9% vs. 11%), dyslipidemia (66.4% vs. 12.1%), DM (78.7% vs. 6.4%), ischemic heart disease (12.1% vs. 1.2%), chronic hepatitis (11.9% vs. 2.9%), and lower CCI (2.31 ± 1.61 vs. 0.35 ± 0.78) (all *p* < 0.001). Regarding the biochemical and physiological profiles, liraglutide users almost had significantly higher values for BMI, aspartate transaminase, alanine transaminase, blood urea nitrogen, creatinine, hemoglobin A1c, and fasting sugar than LA users. However, lower mean arterial pressure was noted in the LA group compared with the liraglutide group (129.2 ± 15.6 vs. 132.5 ± 15.6 mmHg; *p* = 0.018). Concerning the lipid profile, compared with the liraglutide group, the LA group had significantly higher high-density lipoprotein (39.8 ± 9.8 vs. 44.3 ± 8.5 mg/dL; *p* = 0.013) and low-density lipoprotein (104.8 ± 32.7 vs. 125.2 ± 24.3 mg/dL; *p* < 0.001) but lower triglyceride levels (233.3 ± 229.9 vs. 151.8 ± 75.8 mg/dL; *p* < 0.001). We used overlap weighting to eliminate the differences in baseline features between the two groups in terms of age, gender, CCI, and body weight (Table 1).

### 3.2. Assessment of Body Weight Changes

LA users exhibited a two-fold more significant weight loss than liraglutide users after 180 days of intervention (5.82 ± 4.39 vs. 2.37 ± 5.75 kg; *p* < 0.001) (Figure 3A). Throughout the study, there was a notable difference in the pattern of body weight reduction between and within the two groups. Liraglutide users significantly reduced body weight during the first month, whereas LA users experienced a significant reduction during the first 90 days (Figure 3A). The trend of body weight reduction was similar with overlap weighting liraglutide and LA users (Figure 3B). Moreover, LA users had a two-fold greater reduction in BMI than liraglutide users (−2.27 ± 1.73 vs. −0.93 ± 2.25 kg/m^2^ and −2.20 ± 1.07 vs. −0.81 ± 0.66 kg/m^2^, with and without overlap weighting, respectively; all *p* < 0.001) (Table 2).

A significantly higher proportion of LA users achieved the goal of >5% weight loss compared with liraglutide users (64.2% vs. 22.7%; effect size: 6.09, 95% CI: 4.22–8.79; *p* < 0.001). Similar trends were observed for >10% (26.6% vs. 4.2%; effect size: 8.27, 95% CI: 4.87–14.05; *p* < 0.001) and >15% (6.9% vs. 1.2%; effect size: 6.02, 95% CI: 2.33–15.53; *p* < 0.001) weight loss. With overlap weighting, a similar trend in body weight loss was observed in both LA and liraglutide users. Most LA and liraglutide users lost 5–10% and >5% of their body weight, respectively (Figure 4 and Figure 5). Additionally, sensitivity analyses with 1:1 PSM and IPTW were conducted. In 1:1 PSM, LA users lost more body weight than liraglutide users (−5.67 ± 5.16 kg vs. −1.3 ± 3.27 kg, *p* < 0.001). This trend was consistent in the IPTW model, with LA users showing greater weight loss than liraglutide users (−5.92 ± 4.4 kg vs. −2.34 ± 2.97 kg, *p* < 0.001; Appendix A). Interestingly, increases in body weight were noted in both groups during the study period; however, the proportion of subjects who experienced weight gain was markedly lower in the LA group than in the liraglutide group (Figure 5).

### 3.3. Adverse Events

At the end of the trial, 17 liraglutide users had hypertension, and four had ischemic heart disease, while there was no new-onset hypertension or ischemic heart disease observed among LA users (Table 3). With balancing through 1:1 PS M, no significant adverse events were observed during treatment in either group. In the IPTW model, there was no adverse event in the LA users, but a lower risk of hypertension was noted compared to liraglutide users (0% vs. 2.86%, *p*-value = 0.025; Appendix A).

## 4. Discussion

Our study results suggest that LA may be a more effective treatment option than liraglutide use for reducing body weight. After 180 days of treatment, the LA group experienced a mean reduction in body weight of 7.16% (5.82 ± 4.39 kg), which was significantly greater than that recorded for the liraglutide group (2.62%; 2.37 ± 5.75 kg). Given the nature of this cohort study, differences in the covariates between the LA and liraglutide groups were expected, with certain factors such as gender, age, baseline weight, and comorbidities potentially influencing the conclusions (Table 2). To address this, we included these covariates in the PS models to balance the differences between potential confounding variables. The reductions in body weight and BMI varied across the PS models, indicating that, while these covariates may have been influential, their impact on the overall conclusions was minimal. Different PS models utilizing the same covariates yielded consistent results across various weighting and matching conditions. This consistency indicated that the differences observed between or within the groups receiving LA and liraglutides were reliable. Additionally, no adverse events were reported in the LA group, indicating that LA may be a safer option for weight loss than liraglutide. Based on a 2009 randomized controlled trial (RCT) published in *The Lancet* [30], and another RCT published in *The Lancet* in 2017 [31], it has been demonstrated that once-daily subcutaneous liraglutide, when combined with diet and exercise, can reduce body weight by 4.8–6.5 kg. However, subjects with higher initial weights tend to experience a decrease in the efficacy of liraglutide [32], which might explain why the mean weight reduction observed with liraglutide in this observational study was similar to that seen in clinical trials. 

On the other hand, previous studies involving postmenopausal individuals have shown that the combination of LA and a low-calorie diet can lead to a reduction of 9% in body weight after 6 months of treatment [23]. A randomized control trial demonstrated that LA significantly reduced postpartum weight by approximately 19% of the BMI after 36 weeks of treatment; of note, subjects in the sham group experienced only a 3% reduction in BMI [33]. However, based on different study designs, weight and BMI reduction varied significantly across different studies, with weight loss ranging from 3.17 to 16.33 kg and BMI reduction ranging from 1.45 to 6.72 kg/m^2^ [22,23,24,33,34,35]. These discrepancies may be attributed to differences in study populations and the specific LA settings used. Although these studies assessed the effect of LA in conjunction with dietary restrictions and/or exercise, the weight reduction observed in our study (5.82 kg) was less pronounced. This may be due to the observational nature of our study, where only diet education could be provided during routine clinical practice. Rigorous dietary restrictions, exercise protocols, and adherence could not be closely monitored, potentially leading to less substantial weight reduction. Nonetheless, alongside previous studies, our findings suggest the feasibility of using LA for body weight management, highlighting the need for clinical trials to evaluate the effectiveness of our LA settings. 

From the perspective of traditional Chinese medicine, obesity is often attributed to an accumulation of phlegm-dampness. This condition is believed to result from irregular dietary habits or impaired spleen function, leading to the accumulation of dampness, which transforms into phlegm and causes obesity, like lipid deposition in liver or subcutaneous tissues. Therefore, in acupuncture point selection, points along the spleen and stomach meridians, such as ST25, ST28, ST40, and SP15, are chosen to enhance spleen function and promote the elimination of dampness. Additionally, auricular points like stomach and hunger are used to help control appetite and regulate water metabolism.

LA has demonstrated a favorable safety profile and higher effectiveness for weight loss compared with other non-pharmacological interventions. Higher baseline body weight may be associated with greater body weight loss; hence, the lower baseline body weight and greater weight reduction observed in the LA group are intriguing [36]. Additionally, we observed no occurrence of significant systemic adverse events during the study’s observation period. Considering LA’s non-invasive nature, it may become an integrative and complementary treatment for obesity. It has been suggested that non-invasive brain stimulation could assist in the management of obesity, yielding a significant decrease in BMI (4–8%) and few adverse events (e.g., headache) [37,38]. Traditional needle acupuncture could result in moderate reductions in body weight in overweight patients (25 ≤ BMI < 30), with possible side effects including dizziness, nausea, fatigue and local itching, tingling, minor inflammation, or pain at the acupuncture site [39,40]. Tsz Fung Lam et al. reported that electro-acupuncture in treating central obesity could reduce BMI −0.6 (95% CI = −0.9 to −0.3) in 8 weeks with adverse effects such as headaches, dizziness, and insomnia [41]. Invasive body contouring can effectively remove localized areas of adiposity from under the skin. However, the possibility of adverse events (e.g., prolonged swelling, bruising, numbness, thrombophlebitis, and pulmonary embolism) raises concerns [42]. Non-invasive body contouring techniques using high-intensity focused ultrasound have shown promising results in reducing the circumference of the treated area and skin fat thickness. However, of note, this procedure may not be suitable for pregnant women, individuals with a BMI >30, or those with poor medical status [43]. In summary, based on our findings, it can be inferred that LA may exert a significant weight loss effect with good tolerability and minimal harm to individuals.

LA could be used as a non-pharmacologic adjunct therapy to enhance weight loss for individuals receiving pharmacologic therapies, regardless of the potential drug interactions. As a GLP-1 receptor agonist, liraglutide acts on GLP-1 receptors and suppresses appetite in humans [44]. Liraglutide may reduce hunger signals and induce the feeling of fullness by stimulating proopiomelanocortin (POMC) neurons and inhibiting neuropeptide Y (NPY) and agouti-related peptide (AGRP) neurons in the arcuate nucleus [45]. Thus, obese subjects who received GLP-1 experienced a reduction in body weight due to a slower rate of gastric emptying and an extended period of postprandial satiety [46]. The mechanisms underlying the effects of LA on weight reduction and loss of appetite remain unclear [25]. For example, the acupoints on ears may affect motility and the gastrointestinal tone by regulating the vagal nerve [47]. Additionally, from the perspective of traditional Chinese medicine, obese individuals tend to have a phlegm-dampness constitution, which was reported as the main risk factor of metabolic disorder [48]. The acupoints used in the present study aimed to reduce the phlegm-dampness accumulation in the body by regulating qi deficiency and addressing the resulting metabolism disorder [24]. Studies on animals and humans have indicated that acupuncture reduces weight by regulating obesity-related neuropeptides (e.g., ghrelin, leptin, or serotonin) and reducing lipid levels [49,50]. Using functional magnetic resonance imaging, Von Deneen et al. found that acupuncture can also stimulate several neurophysiological pathways, and affected satiety hormones and the basal metabolic rate [51]. Moreover, it has been reported that LA can improve glycemic control and markers of insulin resistance. These effects could be complementary to the effects of GLP-1 agonists on obesity [52]. 

Our results also revealed that 64.2% of LA users achieved a weight loss of 5–10% of their baseline body weight compared with only 22.7% of liraglutide users. Previous studies have shown that a 5–10% weight loss is associated with significant health benefits. These benefits include a reduction in healthcare costs, improvement in glycemic outcomes and quality of life, and reduction in the risk of metabolic diseases, cardiovascular disease, hepatic steatosis, and structural damage to the knee joint [53,54]. It has been reported that a body weight reduction of 10% in patients with type 2 diabetes may lower the risk of acute myocardial infarction, stroke, or admission to hospital for angina by 21% [55]. Studies have demonstrated that weight loss promotes clinical improvement in patients with non-alcoholic fatty liver disease [56,57]. In the present study, the target of 5–10% body weight reduction was achieved. Therefore, LA may help individuals achieve their weight-loss goal and reduce the risk of comorbid diseases.

### Limitations

The present study was subject to some limitations. Firstly, given that this retrospective cohort study was based on patient data retrieved from the hospital database and involved subjects seeking interventions other than diet education or exercise, identifying patients treated with a placebo or those who did not receive any treatment to serve as a control group was challenging. Consequently, we selected liraglutide users as the active control group to emulate real-world situations since liraglutide is one of the most potent FDA-approved medications for weight management. Secondly, we focused on the treatment, diagnosis, and related biochemical and physical parameters of obesity. Therefore, we may have overlooked information related to lifestyle modifications and subject compliance to treatment with LA or liraglutide. However, the real-world nature of this study may provide more realistic insight into the management of obesity. Thirdly, as most subjects were collected at the CGMH, the cohort study was predominantly composed of Asian individuals. It remains unclear whether these results are generalizable to other ethnic groups. Fourthly, the long-term treatment effects of LA remain unknown, as our study tracked the subjects for only 180 days. Fifthly, although the authors have consensus based on the protocol provided by Dr. Wen-Long Hu [24,33], it is true that the doctors may add other acupoints to relieve other symptoms or modify individuals’ constitution when managing body weight. However, it is acknowledged that physicians may add other acupoints to alleviate additional symptoms or adjust the individual’s constitution while managing body weight. The impact of these additional acupoints on weight control remains unclear, as choosing specific acupoints may not be more effective than selecting non-specific ones [58]. Nonetheless, this study demonstrates the effectiveness of applying laser acupuncture to the most crucial acupoints for body weight control. Further clinical studies are required to elucidate this matter fully. Finally, some baseline demographic differences existed between the two groups. Although the PS-based models balanced these differences, larger randomized controlled trials are necessary to confirm the true efficacy of LA on obesity, as well as its potential side effects.

## 5. Conclusions

This study determined that LA may effectively reduce body weight in a real-world setting while causing minimal adverse events or comorbidities. Compared with liraglutide use, LA was linked to a slightly greater reduction in body weight. Owing to its non-invasive nature, good safety profile, and lack of drug–drug interactions, it appears promising to investigate the effects of LA alone or in combination with liraglutide in future studies. Further large-scale clinical trials are necessary to establish the effectiveness and safety of LA, as well as to assess the potential long-term comorbidities and adverse events associated with this treatment.

## Figures and Tables

**Figure 1 healthcare-12-01279-f001:**
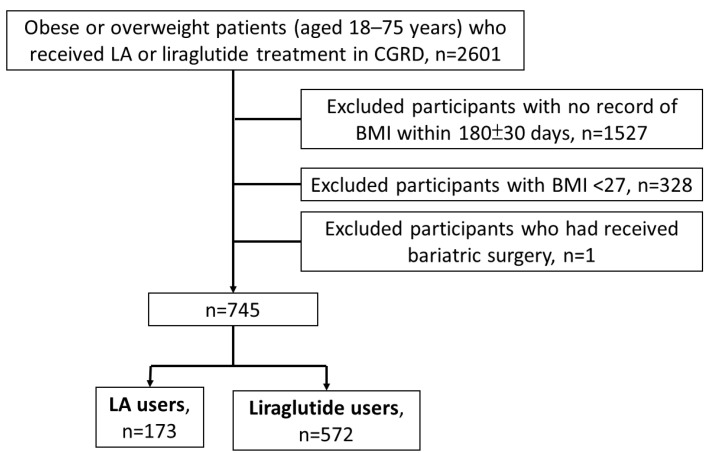
Flow diagram of this study. Abbreviations: CGRD, Chang Gung Research Database; LA, laser acupuncture.

**Figure 2 healthcare-12-01279-f002:**
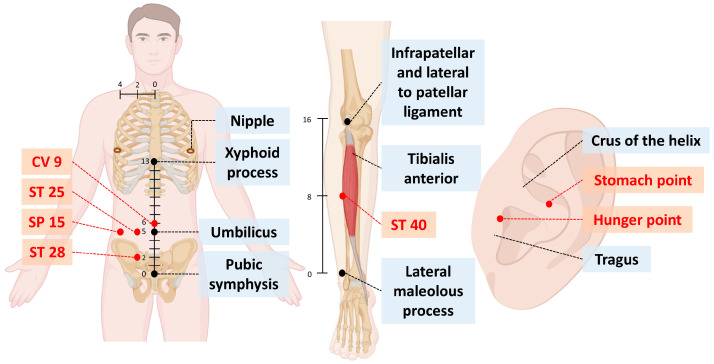
Schematic diagram of acupoints used for laser acupuncture (LA). The unit for precise acupoint localization is the Proportional Bone Cun (B-cun). The following measurements were defined: 8 B-cun for the distance between the two nipples; 5 B-cun for the length from the umbilicus to the superior border of the pubic symphysis; 8 B-cun for the length from the midpoint of the xiphisternal synchondrosis; and 16 B-cun for the length from the prominence of the lateral malleolus to the popliteal crease to the umbilicus (edited with BioRender).

**Figure 3 healthcare-12-01279-f003:**
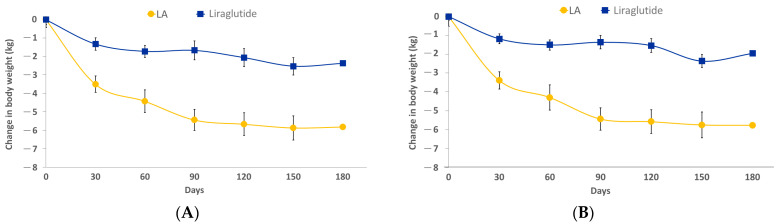
Changes in body weight during the 180 days of treatment (**A**) without overlap weighting (*p*-values < 0.001 for both time and treatment effects; F values for time factor: 19468; and F values for treatment factor: 153633) and (**B**) with overlap weighting (*p*-values < 0.001 for the overall and both the time and treatment effects; F values for time factor: 10460; and F values for treatment factor: 67923). Abbreviations: LA, laser acupuncture.

**Figure 4 healthcare-12-01279-f004:**
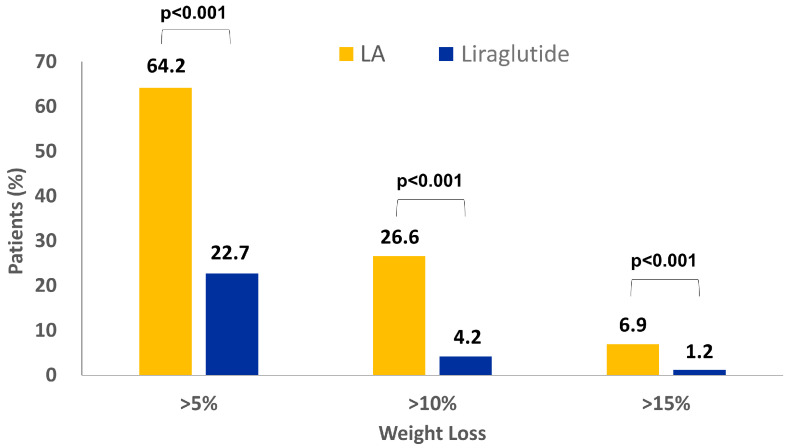
Subjects achieving >5%, >10%, and >15% body weight loss after 180 days of treatment. Abbreviations: LA, laser acupuncture.

**Figure 5 healthcare-12-01279-f005:**
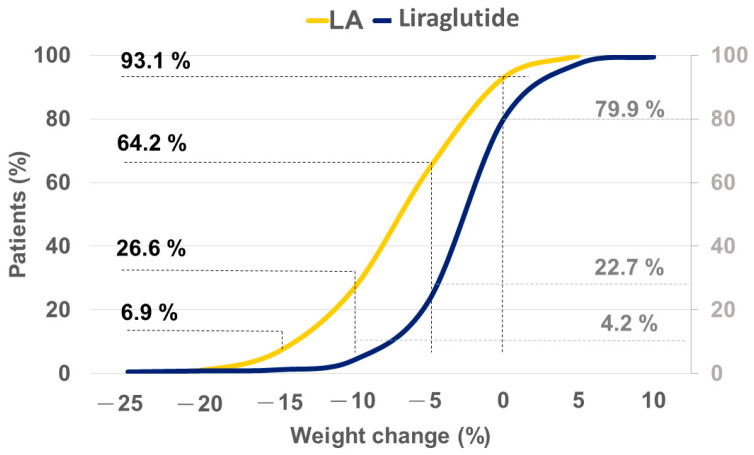
Accumulated distribution of 5%, 10%, and 15% of body weight loss among LA and liraglutide users. Abbreviations: LA, laser acupuncture.

**Table 1 healthcare-12-01279-t001:** Baseline demographic characteristics of LA and liraglutide users.

	LA Users	Liraglutide Users	*p*-Value	Overlap Weighting		ASMD
	(n = 173)	(n = 572)		LA Users	Liraglutide Users	
Demographics						
Age, years	41.7 ± 11.0	51.7 ± 11.7	<0.001	45.4 ± 6.4	45.4 ± 3.9	0.000
Gender (n, %)			<0.001			0.000
Male	28 (16.2%)	282 (49.3%)		26.2%	26.2%	
Female	145 (83.8%)	290 (50.7%)		73.8%	73.8%	
Comorbidities						
Hypertension	19 (11%)	354 (61.9%)	<0.001	21.5%	36.8%	0.343
Dyslipidemia	21 (12.1%)	380 (66.4%)	<0.001	20.4%	50.2%	0.656
Ischemic heart diseases	2 (1.2%)	69 (12.1%)	<0.001	1.7%	6.0%	0.225
CVD	2 (1.2%)	15 (2.6%)	0.386	2.53%	0.05%	0.221
CPD	8 (4.6%)	34 (5.9%)	0.513	10.44%	0.59%	0.442
DM	11 (6.4%)	450 (78.7%)	<0.001	14.7%	67.1%	1.260
Chronic hepatitis	5 (2.9%)	68 (11.9%)	<0.001	7.0%	2.0%	0.242
NAFLD	1 (0.6%)	10 (1.8%)	0.472	0.70%	1.16%	0.048
CCI	0.35 ± 0.78	2.31 ± 1.61	<0.001	0.81 ± 0.66	0.81 ± 0.21	0.000
Biochemical and physiological profiles						
MAP, mmHg	132.5 ± 15.6	129.2 ± 15.6	0.018	135.2 ± 9.5	132.1 ± 5.5	0.399
Body weight, kg	81.4 ± 13.1	87.1 ± 16.9	<0.001	83.9 ± 9.0	83.9 ± 5.2	0.000
BMI, kg/m^2^	31.6 ± 4.8	33.8 ± 6.9	<0.001	32.0 ± 2.7	33.3 ± 2.6	0.501
AST, mg/dL	31.1 ± 15.3	50.3 ± 83.9	0.005	33.9 ± 12.9	44.4 ± 15.5	0.736
ALT, mg/dL	36.3 ± 27.3	49.8 ± 48.7	0.001	41.9 ± 21.4	49.5 ± 13.7	0.421
BUN, mg/dL	12.3 ± 3.1	26.4 ± 19.7	<0.001	12.5 ± 2.3	16.5 ± 2.4	1.667
Creatinine, mg/dL	0.66 ± 0.14	1.13 ± 1.04	<0.001	0.69 ± 0.11	0.81 ± 0.12	1.053
HbA1C, %	6.4 ± 0.8	9.7 ± 1.7	<0.001	6.6 ± 0.7	9.5 ± 0.6	4.802
Fasting glucose, mg/dL	104.2 ± 24.0	214.5 ± 72.6	<0.001	113 ± 18.4	201.2 ± 22.6	4.284
Lipid profile						
Total cholesterol, mg/dL	197.9 ± 30.1	185.8 ± 44.7	0.051	199.9 ± 22.8	191.5 ± 13.4	0.449
Triglyceride, mg/dL	151.8 ± 75.8	233.3 ± 229.9	<0.001	161.4 ± 61.8	224.3 ± 51.8	1.103
LDL cholesterol, mg/dL	125.2 ± 24.3	104.8 ± 32.7	<0.001	125.6 ± 18.7	112.2 ± 11.3	0.869
HDL cholesterol, mg/dL	44.3 ± 8.5	39.8 ± 9.8	0.013	43.8 ± 6.9	40.88 ± 3.2	0.543

Continuous covariates are presented as the mean ± standard deviation, while categorical covariates are presented as a number (percentage). The covariates were balanced using the absolute standardized mean difference (ASMD). An ASMD ≤ 0.1 indicates a negligible difference in potential confounders between the two study groups. Abbreviations: ALT, alanine transaminase; AST, aspartate transaminase; BUN, blood urea nitrogen; CCI, Charlson comorbidity index; CPD, chronic pulmonary disease; CVD, cerebrovascular diseases; DM, diabetes mellitus; HDL, high-density lipoprotein; NAFLD, non-alcoholic fatty liver disease; LA, laser acupuncture; LDL, low-density lipoprotein; MAP, mean arterial pressure.

**Table 2 healthcare-12-01279-t002:** Changes in primary endpoints from baseline to day 180, with and without overlap weighting.

	Without Overlap Weighting			With Overlap Weighting		
	LA Users (n = 173)	Liraglutide Users (n = 572)	t or z, *p*-Value (between Group)	LA Users	Liraglutide Users	t or z, *p*-Value (between Group)
Changes in body weight						
Weight, kg	−5.82 ± 4.39 (−6.48, −5.16)	−2.37 ± 5.75 (−2.84, −1.90)	−7.28, <0.001	−5.77 ± 2.82 (−6.19, −5.35)	−1.95 ± 1.39 (−2.06, −1.84)	−24.14, <0.001
Within group t, *p*-value	7.82, <0.001	3.18, 0.002		8.07, <0.001	8.74, <0.001	
% of body weight	−7.16 ± 0.05 (−7.17, −7.15)	−2.62 ± 0.05 (−2.63, −2.62)	−1045.16, <0.001	−6.83 ± 0.03 (−6.84, −6.83)	−2.33 ± 0.02 (−2.33, −2.33)	−1851.85, <0.001
Loss >5% (%)	64.2	22.7	10.225 <0.001	61.1	19.7	10.482, <0.001
Loss >10% (%)	26.6	4.2	8.8456, <0.001	27.5	5.1	8.493, <0.001
Loss >15% (%)	6.9	1.2	4.1885, <0.001	6.1	1.6	3.232, 0.002
Changes in BMI	−2.27 ± 1.73	−0.93 ± 2.25	−8.29, <0.001	−2.20 ± 1.07	−0.81 ± 0.66	−16.20, <0.001
Within group t, *p*-value	5.86, <0.001	3.05, 0.002		10.00, <0.001	7.34, <0.001	

Continuous covariates are presented as the mean ± standard deviation, while categorical covariates are presented as a number (percentage). The *p*-values were calculated by Student *t*-test and Pearson’s chi-squared test for continuous and categorical covariates, respectively. Paired *t*-tests were used to compare the within-group weight and BMI changes. Abbreviations: BMI, body mass index; LA, laser acupuncture

**Table 3 healthcare-12-01279-t003:** Incidence of identifiable adverse events during the 180 days of treatments.

	LA Users	Liraglutide Users	*p*-Value
	(n = 173)	(n = 572)	
Hypertension	0	17	0.017 *
Ischemic heart diseases	0	4	0.578 *
Hemorrhagic stroke	0	0	-
Ischemic stroke	0	0	-

* Fisher’s exact test. Abbreviations: LA, laser acupuncture.

## Data Availability

The data that support the findings of this study are available on request from the corresponding author.

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
