# Peer review of "Laser Acupuncture versus Liraglutide in Treatment of Obesity: A Multi-Institutional Retrospective Cohort Study"

_healthcare, 2024, doi:10.3390/healthcare12131279_

Round 1
Reviewer 1 Report
Comments and Suggestions for Authors
Methods:
Lines 110-112 "The LA group received LA three to four times per week on fixed acupoints from a qualified practitioner of traditional Chinese medicine. The choice of acupoints and frequency of LA was based on the study conducted in 2010 by Dr. Wen-Long Hu”
In this observational study, how did you control the selection of acupoints for treating obesity?
Results:
Lines 188-189 "LA users exhibited a two-fold more significant weight loss than liraglutide users after 180 days of intervention (5.82±4.39 vs. 2.37±5.75 kg; p<0.001)"
The reported mean change of body weight in the pooled studies estimates showed that liraglutide treatment resulted in -4.91 kg. On the other hand, a few studies with LA reported that LA decreased body weight with a mean of -1.7. Please discuss it.
Author Response
as the attached file

Reviewer 2 Report
Comments and Suggestions for Authors
The authors presented an interesting comparative research between laser acupuncture and liraglutide for obesity treatment. The argument put forward by the authors to justify this study is well sustained through the literature review, alongside comparative techniques aiming for the same therapeutic outcome. However, although the authors used a publicly available and well-structured medical database from CGRD, I am having relevant concerns that would like the authors to address as follows:
1) The study under consideration does not seem to have a control group to statistically validate the conclusions.
2) Although authors argue the contrary, the dataset seems very unbalanced. In line 155, there is a description of a technique used to deal with this problem (Overlap Weighting) that is valid for observational studies that rely on propensity scores. Is this an observational study? Are the authors dealing with propensity scores?
3) I cannot see how the covariates described in Section 2.4 (line 128) influence the overall comparison between treatments whatsoever.
4) Could the authors please clarify how they are using descriptive statistics for performing comparisons? (wouldn’t be inferential statistics instead?) (line 149)
5) This is a longitudinal study. Hence, independent t-test is not the correct statistical technique for the analysis of BMI changes.
6) Following 5) above, chi-square tests for proportions need to be corrected for pairwise comparisons.
7) In line 159, authors claim to have undertaken sensitivity tests. However, I could not find any further discussion as to how these tests affected the interpretation of the results, nor actual metrics seem to be present in the paper.
8) Following 5) and 7) above, the authors did not present actual inferential metrics whatsoever, such as the chi-square statistics, effect sizes, and corresponding confidence intervals. This could potentially invalidate the conclusions achieved.
Comments on the Quality of English Language
No major issues detected.
Round 2
Reviewer 2 Report
Comments and Suggestions for Authors
Please, have a read to the attached document.

No main issues found.
